# Knowledge and Disposal Practice of Leftover and Expired Medicine: A Cross-Sectional Study from Nursing and Pharmacy Students’ Perspectives

**DOI:** 10.3390/ijerph17062068

**Published:** 2020-03-20

**Authors:** Adel Bashatah, Syed Wajid

**Affiliations:** 1Department of Nursing Administration and Education, College of Nursing, King Saud University, Riyadh 11362, Saudi Arabia; 2Department Clinical Pharmacy, Drug and Poison Information Centre, College of Pharmacy, King Saud University, Riyadh 11451, Saudi Arabia; wali@ksu.edu.sa

**Keywords:** Expired, Medicine, Unused, Nurses, Pharmacist, Saudi Arabia

## Abstract

The objective of the present study was to investigate the knowledge and practices concerning unused and expired medicine among pharmacy and nursing students at King Saud University, Riyadh, Saudi Arabia. A cross-sectional study design was used. The study used a validated paper-based, self-administered questionnaire. Statistical analyses were performed using SPSS, Version 24. The response rate was 70.4% (n = 352). The results indicated that 57.4% of pharmacy students and 53.4% of nursing students check the expiry date of medicine before procuring, and 37.6% of pharmacy students and 52.5% of nursing students keep unused medicine until it expires. With regard to disposal, 78.9% of pharmacy students and 80.5% of nursing students reported discarding expired medicine in household garbage or flushing it down a sink or toilet. Only a small percentage returns leftover medicine to a medical store. There was a statistically significant difference between pharmacy and nursing students in regard to checking the expiry date of medicine before procuring (*p* = 0.01), and keeping unused medicine until it expires (*p* = 0.03). The study concluded that the majority of respondents dispose of medicine unsafely. The findings suggest that creating awareness regarding proper medicine disposal procedures among university health care students in Saudi Arabia is needed.

## 1. Introduction

Universally, the correct disposal of medicine poses a challenge and is receiving increasing attention [1,2,3,4,5]. A lack of knowledge about proper disposal of unused medicine can lead to serious consequences, such as an accumulation of toxins and chemicals from the medicine into the environment, unintentional overdose, and prescription drug abuse [6]. A number of countries have established guidelines for disposing of leftover or expired medication [7,8,9,10,11,12].

A study by AlAzm et al in 2017, carried out in King Abdul-Aziz Medical City (KAMC), Jeddah, found that a large majority of subjects (80%) never received any information about the safe disposal of medication from their health care providers; 73% discarded leftover medication in the trash, while 14% returned it to a pharmacy [13]. A study in 2003 investigating the use and disposal of medication among Saudi families reported that most of the subjects stored or discarded unused medicine [14]. Fletcher found that a lack of education, older age, and receiving multiple medications were factors associated with inappropriate drug use, which could lead to medication waste in the environment [15].

Reports have confirmed a lack of knowledge among health care professionals about the significant risks of unused or expired medicine, including diversion, abuse, and accidental overdose [16]. A study by Raja et al among health care professionals reported partial knowledge about proper drug disposal and a lack of safe disposal practices [5].

The literature has also shown that health care students acquire information about drug use and disposal from a range of sources, including health care practitioners, curricula, media, teachers, and medicine package inserts. Knowledge about proper drug practice formed prior to graduation may affect medicine disposal practices later during employment [5].

The clinical benefits resulting from collaboration among health care professionals, such as pharmacists and nurses, have been proven by many studies [12,17,18]. Nurse practitioner and pharmacist consultations in family practice resulted in major improvements in the correct use of medication [5,17]. Fletcher reported that nurse practitioner and pharmacist counseling is important in improving the appropriate use of medication [15].

While a number of studies exist about safe drug disposal knowledge and practice, much of the early work is limited to the general public, both in Saudi Arabia and in other countries [2,3,4,5]. There is a shortage of research on the knowledge and practice of drug disposal among health care students, such as pharmacy and nursing students, who are on the ‘front lines’ in the health care setting. A review of the literature found no studies carried out in Saudi Arabia that investigated pharmacy and nursing students’ knowledge and disposal practices regarding unused and expired medicine. Therefore, the present study, conducted among pharmacy and nursing students from King Saud University Colleges of Pharmacy and Nursing in Saudi Arabia, will add to the investigative literature.

## 2. Methods

### 2.1. Study Design, Area, and Setting

This study was conducted for a period of 3 months, from January to March 2019, at the College of Pharmacy and the College of Nursing, King Saud University, Riyadh, Saudi Arabia. The study included senior-level students of nursing and pharmacy (4th, 5th and 6th year pharmacy and nursing students). Data collection was carried out with a paper-based, printed questionnaire through a self-administration procedure using convenience sampling. Convenience sampling is a non-probability technique in which study subjects are selected based on certain criteria, such as availability at a given time, willingness to participate, accessibility, and geographical proximity to the researchers [19]. The study was approved by the ethics committee of the College of Medicine at King Saud University (E-19-4279).

The study tool was prepared after conducting an extensive review of questionnaires used in previous studies that evaluated the disposal of expired and unused medication [4,5]. Reliability of the questionnaire was assessed in a pilot study conducted among a sample of 10 randomly selected pharmacy and nursing students. Cronbach’s alpha coefficient was 0.70, which indicated that the questionnaire could be used in this study. Slight modifications to the survey language were made, and the questionnaire was amended to be suitable for Saudi students in Saudi Arabia. The questionnaire consisted of 10 items divided into two parts and included both multiple choice and binary answers. Part A contained demographic questions, including age, marital status, and methods for obtaining medication. Part B addressed students’ knowledge and habits regarding unused and expired medicine.

### 2.2. Data Collection Procedure

The study population included nursing and pharmacy students who were regular students at the colleges and willing to participate in the survey. A researcher who was appointed to collect the data visited the students in their classrooms during lecture periods. A brief talk was given to explain the purpose of study and to ensure the students of confidentiality in their responses. Written informed consent was obtained from the students. Participants were given enough time to complete the questionnaire. Data collection was conducted using convenience sampling and performed in such a way that all of the pharmacy and nursing students would be included. Students with incomplete answers to more than a half of the study questionnaires were considered as incomplete responses and therefore excluded from the study, and students who missed 2 or 3 questions in the survey were considered as a treatable response and therefore included in the study. Students who did not return questionnaires were considered non-respondents.

### 2.3. Statistical Analyses

The data were entered and coded, and descriptive statistics were calculated for all survey items. All statistical analyses were conducted using SPSS Version 24 (SPSS Inc., Chicago, IL, USA). The results are expressed as numbers and percentages presented in the forms of tables and graphs. In addition, the associations between variables were determined by performing chi-square tests. A *p*-value < 0.05 was considered a statistically significant difference in all analyses.

## 3. Results

In total, 500 questionnaires were distributed during the study period, and the response rate was (n = 352, 70.4%). Of all participating subjects, 161 (45.3%) were pharmacy students, and 191 (54.6%) were nursing students. The majority of students (92.6%) were single. Most of the respondents were between 18 and 22 years of age. Nearly 59.2% of pharmacy students and 70.5% nursing students reported that they were storing unused medicine in their home. Slightly more than half (51.5%) of pharmacy students, and 36.1% of nursing students purchased medicine over the counter. Details are presented in Table 1.

Of the participants, 57.4% of pharmacy students and 53.4% of nursing students reported that they check the expiry date of medicine before procuring it from the pharmacy, while 22.8% of pharmacy students and 35.6% of nursing students said they do not check it. A small percentage of both groups (18% and 9.5%, respectively) said they did not know.

Nearly half of pharmacy students (47.2%) and over half of nursing students (61.2%) threw away leftover medicines in household garbage, while 6.8% and 5.3%, respectively, flushed unused medicine down the sink or toilet. A large majority of both pharmacy (68.3%) and nursing (74.2%) students said they discard expired medication in household garbage, while 10.6% and 6.3%, respectively, flushed expired medicine down the toilet or sink. Interestingly, only a small percentage of both groups said they return leftover or expired medicine to the medical store or pharmacy.

More than half (64.6%) of pharmacy students and a majority (81.6%) of nursing students agreed that the responsibility for creating awareness of proper disposal methods for leftover and expired medicine lies with the Ministry of Health. Additionally, 19.9% and 16.3% of pharmacy and nursing students, respectively, believed that pharmacists bear the responsibility for this awareness. Large majorities of both groups (91.9% and 81.8%) accepted that inappropriate disposal of unused and expired medicine can affect the environment and health. Detailed information is presented in Table 2.

The most common classess of drugs purchased by respondents were painkillers, followed by antibiotics. The detailed information was given in Figure 1. There was a statistically significant difference between pharmacy and nursing students regarding checking the expiry date of medicine before procuring (*p* = 0.010), unused medicine being stored at home (*p* = 0.030), and awareness that improper disposal can affect the environment (*p* = 0.007). There was no statistically significant difference in the belief that the pharmaceutical industry and the general public are responsible for creating awareness of proper disposal of unused and expired medicines (< 0.05). Details are presented in Table 3.

## 4. Discussion

Globally, the improper disposal of leftover medicine poses a danger to public health and environmental safety [20,21]. Therefore, it is important to investigate the prevalence of this practice and to create awareness of and solutions for safe medicine disposal methods, particularly among health care providers. This is the first study of its kind to survey university health care students in Saudi Arabia on their knowledge and practice regarding unused and expired medication. The present study results show that majority of the participants prefer to purchase medicines using a prescription, which is similar to the previous study conducted among health care students and staff, in which nearly 55% of surveyed participants purchased medicine through prescription [5]. It is believed that health care professionals such as pharmacist and nurses work through interprofessional collaboration to dispense medications to the patients. Additionally, a number of studies revealed that nurses and pharmacists provide counseling and education regarding the use and administration of drugs, and to emphasize the importance of expiry dates to patients [5,19].

The results show that a majority of pharmacy and nursing students (58.2% and 54%, respectively) check the expiry date of medicine before purchasing it. These results are lower than the findings of a study by Raja et al., in which 98% of nursing students checked the expiry date of medication [5]. Studies have shown that nurses and pharmacists provide counseling and education to patients about the use of medicine and the importance of paying attention to expiration dates [5,20]. In the present study, both pharmacy (68.3%) and nursing (74.2%) students threw away expired medicine in household garbage. These results are in line with those of Raja et al., in which 72% of respondents threw expired medicine in the garbage [5]. A similar study among dental students by Aditya found that 94% of students threw away unused medicine in the household trash [22]. This practice has been observed internationally, irrespective of whether the subjects are the general public or health care professionals [4,5].

In the current study, more than half (52%) of nursing students and 37% of pharmacy students stored leftover medicine. According to Raja et al., the majority of nursing students discarded leftover medicine [5], while studies done among the general public showed that leftover medicine was stored until it expired [17,22,23]. This could be due to the perception that it might be needed in the future [7,22]. However, some studies reported that keeping leftover or unused medicine for a longer time might result in polypharmacy or unintentional consumption of medicine, which can lead to toxic effects in the individuals [4,5,22]. The results also show significant differences between pharmacy and nursing students regarding ways of purchasing medicine (*p* = 0.001). Additionally, results revealed that nursing students were more knowledgeable (*p* = 0.001) in regards to purchasing medicine, checking the expiry date of medicine, and safe disposal practice, compared to pharmacy students. This might be due to the training and clinical rotation, which starts in nursing from the start of the course, while in pharmacy, clinical rotation begins in the last or senior levels.

The majority of the participants in this study said they were aware of the dangers to the environment from improper drug disposal. This finding is similar to those of other studies among both health care students and the general public [5,17,22,23,24,25]. Nevertheless, a general belief persists that flushing unused and expired medication down the toilet or sink is the best practice, particularly for liquid medications [18,25,26,27], even in countries such as the United States and the United Kingdom [26,27].

The present study has several limitations. Firstly, the sample referred to in this study was small, and the research was limited to one university in Riyadh City. Therefore, the outcomes of the current findings can only represent the situation in the current study’s settings. In addition, to address sampling bias due to respondents who did not respond to the survey, it is recommended that future studies investigate the factors or reasons that prevented them from participating.

## 5. Recommendations

It is necessary and advisable to increase education among health care students and the general public about proper storage and disposal of medication, both prescribed and over-the-counter. In Saudi Arabia and other countries, awareness on this issue is lacking. This can be accomplished by the government, learning institutions, and in pharmacies at the point of sale.

It is recommended that the Ministry of Health issue official guidelines and policies for the safe storage and disposal of unused and expired medication. It is also suggested that the government establish a take-back program in partnership with medical stores and the pharmaceutical industry.

Physicians should avoid overprescribing, as this will minimize the burden of unused medication requiring disposal. All of these approaches can play a significant role in reducing the hazards that medicine disposal poses to environmental safety and human health [28].

## 6. Conclusions

The results of this study demonstrate the need for improvement in practices by Saudi health care students regarding the disposal of leftover and expired medicine. Improper disposal of medication can cause contamination of water supplies, introduce toxins into the environment, and risk unintentional overdose or drug abuse. The government should issue guidelines and launch education programs on correct disposal methods for professionals in health care settings and for the general public, as well as establish a convenient medicine take-back program. Future studies can be conducted once these efforts are put in place to monitor their success.

## Figures and Tables

**Figure 1 ijerph-17-02068-f001:**
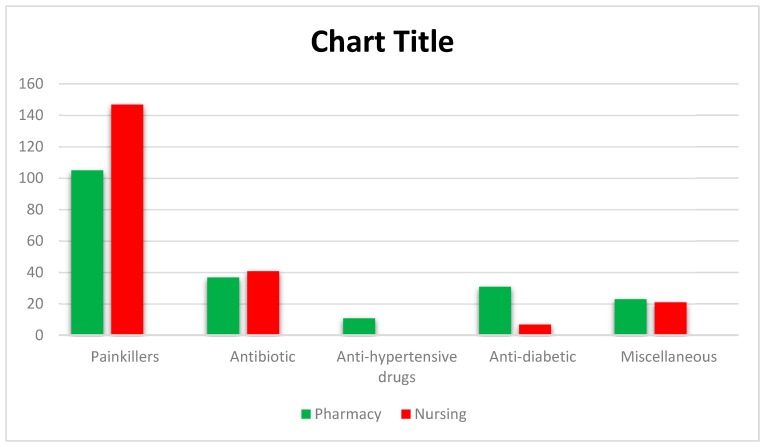
Most common classes of drugs purchased by respondents.

**Table 1 ijerph-17-02068-t001:** Demographics, storage, and procurement of medicine.

Participant Classifications	Pharmacy (n = 161)		Nursing (n = 191)		*p*-Value
	N	%	N	%	
**Age (years)**					
18–22	113	70.8	169	88.8	*p* < 0.001
23–26	46	28.7	14	7.2	
27–30	2	1.2	8	4.1	
**Marital status**					
Single	147	91.3	179	93.7	*p* < 0.05
Married	14	8.7	12	6.3	
**Does any quantity of purchased medicine remain unused in your home *?**					
Yes	96	59.2	134	70.5	*p* < 0.001
No	64	39.5	55	28.9	
**Ways of purchasing medicine**					
Purchased using a prescription	117	72.7	170	89.5	
Purchased without a prescription (OTC)	83	51.5	69	36.1	
Received from a friend /colleague	23	14.8	15	7.9	*p* < 0.001
Obtained / purchased on the advice of a relative	27	16.7	31	16.3	

* Missing responses.

**Table 2 ijerph-17-02068-t002:** Knowledge and practice regarding unused and expired medicine among Saudi pharmacy and nursing students.

Characteristics	Pharmacy n(%)	Nursing n(%)
**Do you check the expiry date of medicine before procuring? ***		
Yes	92 (57.4)	102 (53.4)
No	37 (22.8)	68 (35.6)
Don not know	29 (18)	18 (9.5)
**What do you do with unused medicine?**		
Throw away in household garbage	77(47.2)	117(61.2)
Donate to hospital	18 (11.2)	9 (4.7)
Give to friends or relatives	25 (15.5)	39 (20.5)
Return to medical store	11 (6.8)	9 (4.7)
Keep at home until expired	60 (37.6)	100 (52.5)
Flush in toilet or sink	11 (6.8)	10 (5.3)
**What do you do with expired medicine?**		
Throw away in household garbage	110 (68.3)	141 (74.2)
Flush in toilet or sink	17 (10.6)	12 (6.3)
Give to friends or relatives	8 (5.0)	4 (2.1)
Return to medical store	13 (8.1)	5 (2.6)
Don not know	13 (8.1)	27 (14.2)
**Who is responsible for creating awareness of the proper disposal of unused and expired medicine?**		
Ministry of Health	104 (64.6)	155 (81.6)
Pharmaceutical industry	19 (11.8)	20 (10.5)
Pharmacist	32 (19.9)	31 (16.3)
General public	21 (13)	28 (14.7)
**Improper disposal of unused and expired medicine can affect the environment and health.***		
Yes	132 (81.8)	174 (91.9)
No	29 (18.1)	16 (8.7)

* Missing responses.

**Table 3 ijerph-17-02068-t003:** Association between study participants’ responses in selected variables.

Variables	Pharmacy n(%)	Nursing n(%)	*p*-Value
**Do you check the expiry date of the medicine before procuring?**			
Yes	92 (58.2)	102 (54)	*p* < 0.010
No	37 (23.4)	68 (36.)	
Don not know	29 (18.4)	19 (10.1)	
**Does any quantity of purchased medicine remain unused in your home?**			
Yes	96 (60)	135 (71.1)	*p* < 0.030
No	64 (40)	55 (28.9)	
**Improper disposal of unused and expired medicine can affect the environment and health.**			
Yes	132 (82)	174 (91.6)	*p* < 0.007
No	29 (18)	16 (8.4)	
**Who is responsible for creating awareness of the proper disposal of unused and expired medicine?** Ministry of Health			
Yes	104 (64.6)	156 (82.1)	*p* < 0.001
No	57 (35.4)	34 (17.9)	
Pharmaceutical industry			
Yes	19 (11.8)	20 (10.5)	*p* < 0.701
No	142 (88.2)	170 (89.5)	
Pharmacist			
Yes	32 (19.9)	31 (16.3)	*p* < 0.387
No	129 (80.1)	159 (83.7)	
General public			
Yes	21 (13)	28 (14.7)	*p* < 0.648
No	140 (87)	162 (85.3)	

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
