# Peer review of "Knowledge and Disposal Practice of Leftover and Expired Medicine: A Cross-Sectional Study from Nursing and Pharmacy Students’ Perspectives"

_ijerph, 2020, doi:10.3390/ijerph17062068_

Round 1
Reviewer 1 Report
The manuscript is presenting an interesting topic. The researchers suggest that it is important to create awareness through dispensing policies and procedures.
In general, grammar needs to be reviewed. Verbs are missing from numerous sentences e.g. page 1 lines 17 – 18; page 2 lines 57 – 58; page 6 lines 132-133. The discussion section required extensive grammar editing.
Introduction is lacking numerous key references regarding disposal of medicinal products. Literature review is too short and is not clearly justifying the study or highlighting the significance of the study.
Page 1 line 37 “serious consequences”: please elaborate.
Page 1 lines 36 – 41 the paragraph is vague and is not clearly explaining the importance of knowledge of healthcare students of appropriate disposal of medicines
Methods: Page 2 line 56: what do you mean by senior students? What were the inclusion and exclusion criteria? What was the context in which the questionnaire was previously used in previous literature? Please provide info on data storage and security.
Discussion: Page 6 lines 132-133: please elaborate.
Results need to be critically appraised. Discussion need to reflect the limitations of the study, significance of the findings and implications in practice.
Minor issues: missing spaces e.g. page 2 line 46
Please add a zero before the decimal points for P values e.g. page 3 lines 124-16
Page 6 line 132 disposal
References required extensive editing.
Author Response
Dear editor in cheif , very good afternoon , firstly we thanks to your team for getting back with the comments and giving us chance to publish with you , we will be happy and more happy to publish all our ressearch in your journal . please we sincearly beg you apologise for the errors , accept our manuscript , as we follwoed all the comments from your reviewrs team
In general, grammar needs to be reviewed. Verbs are missing from numerous sentences e.g. page 1 lines 17 – 18; page 2 lines 57 – 58; page 6 lines 132-133. The discussion section required extensive grammar editing.
Introduction is lacking numerous key references regarding disposal of medicinal products. Literature review is too short and is not clearly justifying the study or highlighting the significance of the study.
Page 1 line 37 “serious consequences”: please elaborate.
Answer by author : apologies for the erros , I have made it correct
Page 1 lines 36 – 41 the paragraph is vague and is not clearly explaining the importance of knowledge of healthcare students of appropriate disposal of medicines.
Answer by author: apologies for the erros , I have made it correct
Methods: Page 2 line 56: what do you mean by senior students? What were the inclusion and exclusion criteria? What was the context in which the questionnaire was previously used in previous literature? Please provide info on data storage and security.
Answer by author : apologies for the erros . here we included senior students ( 3rd and fourth years and 5 the years of pharmD and nursing students were included . junior students , like first year second year students from both college of nursing and pharmacy were excluded .
We have collected all the data and extracted for the correctness , to avoid bias in inclusion of the study criteria , data collected was entered in to spss , and kept with us .
Discussion: Page 6 lines 132-133: please elaborate.
Results need to be critically appraised. Discussion need to reflect the limitations of the study, significance of the findings and implications in practice.
Minor issues: missing spaces e.g. page 2 line 46
Please add a zero before the decimal points for P values e.g. page 3 lines 124-16
Page 6 line 132 disposal
References required extensive editing.
my aplogies we have corrected all
thanks

Reviewer 2 Report
The manuscript of Bashatah et al. has performed a cross sectional study among students from pharmacy and nursing college of King Saud University to investigate the trends and knowledge of disposing leftover and expired medications. However, this paper has several major flaws, which is not suitable to be accepted for publication.
- The English language in the text is very hard to follow, as well as many grammar mistakes. It is recommended that the author either ask a colleague whose native language is English to review the manuscript or use language editing services. It will impede the readers to understand what the author is really trying to convey.
- In the title, it is claimed that the trends will also be investigated. However, none of the longitudinal data were collected from the paper.
- The statistical methods should be added in the section of methods. Especially, for the association analysis, the multivariate logistical analysis should be performed (Table 3).
- The non-probability sampling is cheap and easy to perform, but inevitable bias will be brought in. What the author did for bias correction should be stated.
- Only 70.4% of the participants responded to the questionnaire, this seemed very high considering this is a non-probability sampling method. The author should do some further research to understand the underlying reason, incase extra bias was introduced.
- In table 1, if no one was over 31, that row should be removed. In addition, the hypothesis for the statistical test should be clearly stated.
- Considering replacing the word “procuring” if possible, since it is somehow misleading.
- Legend of Figure 1 should be more clear and detailed.
- Line 168, it reads “flushing the expired medications in toilet or sink, is safe practice for liquid medications, reported by many studies.” The author should provide literatures to support this claim, since it violates the common sense.
Author Response
dear editor in cheif very good afternoon
my aplogies for the errors , and we have made all changes and submitted as well for the english gramer editing and we are now submitting the corrected one
i hope it will get accepted
Reviewer 2
The manuscript of Bashatah et al. has performed a cross sectional study among students from pharmacy and nursing college of King Saud University to investigate the trends and knowledge of disposing leftover and expired medications. However, this paper has several major flaws, which is not suitable to be accepted for publication.
- The English language in the text is very hard to follow, as well as many grammar mistakes. It is recommended that the author either ask a colleague whose native language is English to review the manuscript or use language editing services. It will impede the readers to understand what the author is really trying to convey.
Thank you for the revieing our research paper , we will be glad to publish in your journal , also my sincear apologies for the error . we have submitted this paper to enlish and gramer edit as per your comments .
- The statistical methods should be added in the section of methods. Especially, for the association analysis, the multivariate logistical analysis should be performed (Table 3).
Apologies for the error , I have added it . my request , we have performed chisquare test to find association between the variables . so both regression and chisuare will be similar in findings , but methods might be different , we have thought to performed chisuare , so we did it . apologies for this , if still it is the major correction , then we can ask our one of the senior to help us to do logical regression.
- The non-probability sampling is cheap and easy to perform, but inevitable bias will be brought in. What the author did for bias correction should be stated.
We have used The non-probability sampling to achieve a good response rate and targeted ,mainly 3 , 4 and 5 th years of pharmacy nursing students , and we have look carefully about the students and their years of study , we excluded students other than these years of students . we look very carefully in each and every questionnaire to avoid bias.
- Only 70.4% of the participants responded to the questionnaire, this seemed very high considering this is a non-probability sampling method. The author should do some further research to understand the underlying reason, incase extra bias was introduced.
We did this study during the students free time at their class room , we visited 3 rd years students and 4 and 5 th years students separately and on separate week days during their working hours in lecturer , and we included only those students who were present right during the data collection , this method was also folwled among nursing students . that might be the reason we got good response rate and also we explain the importence of study and partipation in the study , we told participations is mandatory
- In table 1, if no one was over 31, that row should be removed. In addition, the hypothesis for the statistical test should be clearly stated.
My apologies for this we have corrected it
- Considering replacing the word “procuring” if possible, since it is somehow misleading.
- My apologies for this we have corrected it
- Legend of Figure 1 should be more clear and detailed.
My apologies for this we have corrected it
- Line 168, it reads “flushing the expired medications in toilet or sink, is safe practice for liquid medications, reported by many studies.” The author should provide literatures to support this claim, since it violates the common sense.
We have did this through an extensive literature am adding here the reference for this
Additionally, it was believed that flushing the expired medications in toilet or sink, is best practice particularly for liquid medications, reported by many studies [24-27].However similar practice was observed in developed countries such as United States and United Kingdom [25-27]. All these approaches considered as the widely used practice for leftover pharmaceuticals, which play significant role in reducing the occurrence of medicine to the environment since it can cause environmental, human health, and safety hazards [27,28].
AlAzmi A, AlHamdan H, Abualezz R, Bahadig F, Abonofal N, Osman M. Patients’ knowledge and attitude toward the disposal of medications. Journal of pharmaceutics. 2017;2017.
Abou-Auda HS. An economic assessment of the extent of medication use and wastage among families in Saudi Arabia and Arabian Gulf countries. Clinical therapeutics. 2003 Apr 1;25(4):1276-92.

Round 2
Reviewer 1 Report
Thanks for the authors for improving the manuscript and making the necessary amendments.
Here are a few more suggestions:
Abstract: Line 18, also in lines 24 and 38: I wouldn’t use the word proper. Are there Saudi evidence-based guidelines or recommended national procedures?
Introduction: lines 35-42, can you please elaborate on current content in undergraduate pharmacy and nurses curricula that is related to disposal of leftover and expired medicines.
Lines 43 and 44: what is the benefit of this collaboration?
Lines 44-47: is this also related to appropriate disposal and handling of medications? Please confirm from the literature.
Line 49: what other countries? Middle East or European as well?
Methods:
Line 61: please define senior students in text between brackets
Line 104: please use ‘expiry date’ instead of ‘expiration’ (please amend in manuscript)
Line 106: please use ‘did not’
Line 112: what is the medical store? Is that a pharmacy?
Line 117: please use ‘inappropriate’ instead of ‘improper’
Table 2: please explain what you have done with missing responses. Please include this in methods
Need to be careful about the generalisability of results as the study concerns one university in one country.
Figure 1: please improve using the journal style with high resolution
Discussion: the discussion has not improved and is still superficial in the way the results are discussed. Results need to be discussed in depth, discussing the differences in the knowledge and disposal practice between nurses and pharmacists, and the impact of demographic aspects, etc.
Was Ethics approval obtained for this study? If not, why not?
Need to discuss limitations of the study
Review of references
Author Response
Dear editor in cheif very good afternoon , my sincear apologise for the delay in submition , we hope our paper will published very soon
Comments and answers
Abstract: Line 18, also in lines 24 and 38: I wouldn’t use the word proper. Are there Saudi evidence-based guidelines or recommended national procedures?
Answer by author
Dear editor and your team , thank you for the comments and my apologies for the erros , in these sentence we were talked about recommended national and international procedures for the proper disposal practice . (USFDA. How to dispose of unused medicines. 2013 [updated 2020 Feb 26; cited 2020 Feb 26]. http://www.fda.gov/ForConsumers/ConsumerUpdates/ucm101653.htm).
Introduction: lines 35-42, can you please elaborate on current content in undergraduate pharmacy and nurses curricula that is related to disposal of leftover and expired medicines
Thank you for the comment , but hear the sentence talking about students , they will get information about safe disposal practice through , their teachers , peers and leaflets , brouchers , but there is no standard guidelines and topics in their graduation syllabus. Most of the students learn disposal practice during their internship form the peers and literature which published in standard national and internation journal . (Raja et al 2018)
Lines 43 and 44: what is the benefit of this collaboration?
Answer by author
Work in collaboration with pharmacist and nurses leads to improvement in clinical practice through , on the time of drug administration , importance of safe dosage administration , also nurses and pharmacist guide patients about quantity of administration , problems with over dosage , storage and disposal , which all leads good medication practice and improved clinilcal benefits .
Lines 44-47: is this also related to appropriate disposal and handling of medications? Please confirm from the literature.
Yes it is related to handling and good disposal practice .
Line 49: what other countries? Middle East or European as well?
Yes it means Europe and asian country
Methods:
Line 61: please define senior students in text between brackets
We defined , thank you .
Line 104: please use ‘expiry date’ instead of ‘expiration’ (please amend in manuscript)
Thank you we did changes
Line 106: please use ‘did not’
Thank you we did changes
Line 112: what is the medical store? Is that a pharmacy?
Of course it is
Line 117: please use ‘inappropriate’ instead of ‘improper’
Thank you we did changes
Table 2: please explain what you have done with missing responses. Please include this in methods
Thank you we did changes
Need to be careful about the generalisability of results as the study concerns one university in one country.
Thank very much , we looked very carefully and done changes.
Figure 1: please improve using the journal style with high resolution
Discussion: the discussion has not improved and is still superficial in the way the results are discussed. Results need to be discussed in depth, discussing the differences in the knowledge and disposal practice between nurses and pharmacists, and the impact of demographic aspects, etc.
Thank you very much for the comments and we did change as per the reviewers recommendation.
Was Ethics approval obtained for this study? If not, why not?
Yes ethical approval was obtained and reference number was given in methods (The study was approved by the ethics committee of the College of Medicine at King Saud University (E-19-4279).
Need to discuss limitations of the study
We did thank you
(The present study has several limitations. Firstly the sample referred in this study was small, and the research was limited to one university in Riyadh city. Therefore, the outcomes of the current findings can only represent the situation in the current study settings only. In addition, to address sampling bias due to respondents who did not respond to the survey, it is recommended that future studies investigate the factors or reasons that prevented them from participating)
Review of references
We did thank you

Reviewer 2 Report
I think the paper has been substantially improved, and is ready to be accepted.
Author Response
Dear editor thank you very much and thank you for all your team members for reviewing our study , i hope it will get accepted